# Sepsis endotypes identified by host gene expression across global cohorts
Josh G. Chenoweth [1] ✉, Joost Brandsma[1], Deborah A. Striegel[1], Pavol Genzor [1], Elizabeth Chiyka[1], Paul W. Blair[1], Subramaniam Krishnan[1], Elliot Dogbe[2], Isaac Boakye[3], Gary B. Fogel[4], Ephraim L. Tsalik [5,12], Christopher W. Woods [5], Alex Owusu-Ofori[2,6], Chris Oppong[7], George Oduro[7], Te Vantha[8], Andrew G. Letizia[9], Charmagne G. Beckett[10], Kevin L. Schully[11] & Danielle V. Clark[1]

## Abstract

**Background** Sepsis from infection is a global health priority and clinical trials have failed to deliver effective therapeutic interventions. To address complicating heterogeneity in sepsis pathobiology, and improve outcomes, promising precision medicine approaches are helping identify disease endotypes, however, they require a more complete definition of sepsis subgroups.

**Methods** Here, we use RNA sequencing from peripheral blood to interrogate the host response to sepsis from participants in a global observational study carried out in West Africa, Southeast Asia, and North America (N = 494).

**Results** We identify four sepsis subtypes differentiated by 28-day mortality. A low mortality *immunocompetent* group is specified by features that describe the adaptive immune system. In contrast, the three high mortality groups show elevated clinical severity consistent with multiple organ dysfunction. The *immunosuppressed* group members show signs of a dysfunctional immune response, the *acute-inflammation* group is set apart by molecular features of the innate immune response, while the *immunometabolic* group is characterized by metabolic pathways such as heme biosynthesis.

**Conclusions** Our analysis reveals details of molecular endotypes in sepsis that support immunotherapeutic interventions and identifies biomarkers that predict outcomes in these groups.

## Plain language summary

Sepsis is a life-threatening multi-organ failure caused by the body's immune response to infection. Clinical symptoms of sepsis vary from one person to another likely due to differences in host factors, infecting pathogen, and comorbidities. This difference in clinical symptoms may contribute to the lack of effective interventions for sepsis. Therefore, approaches tailored to targeting groups of patients who present similarly are of great interest. This study analysed a large group of sepsis patients with diverse symptoms using laboratory markers and mathematical analysis. We report four patient groups that differ by risk of death and immune response profile. Targeting these defined groups with tailored interventions presents an exciting opportunity to improve the health outcomes of patients with sepsis.

Sepsis is life-threatening organ dysfunction due to a dysregulated host response to infection[1]. The signs and symptoms of sepsis are highly variable and typically nonspecific, including aberrations in vital signs like tachycardia and tachypnea alongside indications of organ dysfunction, such as altered mental status, hypotension, and renal dysfunction, potentially leading to shock and death. The infecting pathogen, pathogen load, site of infection, comorbidities, and immunological response by the patient contribute to the clinical presentation and course of disease resulting in a complex heterogeneous syndrome. In addition, the model for host response to sepsis has evolved from a simplified linear hyperinflammatory phase followed by a compensatory anti-inflammatory phase to a more complex paradigm where pro- and anti-inflammatory mechanisms are acting in concert[2]. The failure to develop effective therapeutics and the limited success in developing diagnostic or prognostic tests are often attributed to this clinical and immunological heterogeneity[3,4].

[1]Austere environments Consortium for Enhanced Sepsis Outcomes (ACESO), The Henry M. Jackson Foundation for the Advancement of Military Medicine, Inc., Bethesda, MD, USA. [2]Laboratory Services Directorate, KATH, Kumasi, Ghana. [3]Research and Development Unit, KATH, Kumasi, Ghana. [4]Natural Selection, Inc., San Diego, CA, USA. [5]Center for Infectious Disease Diagnostics and Innovation, Department of Medicine, Duke University School of Medicine, Durham, NC, USA. [6]Department of Clinical Microbiology, Kwame Nkrumah University of Science and Technology (KNUST), Kumasi, Ghana. [7]Accident and Emergency Department, KATH, Kumasi, Ghana. [8]Takeo Provincial Referral Hospital, Takeo, Cambodia. [9]Naval Medical Research Unit EURAFCENT Ghana detachment, Accra, Ghana. [10]Naval Medical Research Command Infectious Diseases Directorate, Silver Spring, MD, USA. [11]Austere environments Consortium for Enhanced Sepsis Outcomes (ACESO), Biological Defense Research Directorate, Naval Medical Research Command-Frederick, Ft. Detrick, Maryland, MD, USA. [12]Present address: Danaher Diagnostics, Washington, DC, USA. ✉e-mail: jchenoweth@aceso-sepsis.org

To overcome the problem of heterogeneity in sepsis there is a growing interest in leveraging biologically related host response phenotypes or *endotypes* to guide precision medicine and companion biomarker discovery for diagnosis and clinical management. Notably, the human host response to infection is highly sensitive and specific, enabling the use of protein and gene expression measurements in peripheral blood for characterization of patient groups based on related underlying sepsis pathophysiology[5–8]. Host gene expression analysis from peripheral blood leukocytes can also diagnose pathogen class or identify subjects that respond differentially to corticosteroid treatment[9–11]. In some cases, these findings have been translated to FDA-approved assays for infectious sepsis diagnosis and pathogenesis[12,13].

Despite these advances, the performance of host-based measures can be population-specific and impacted by the epidemiology of illness[14]. Sepsis is a global health priority accounting for 47 million deaths in 2017. There is an urgent need to identify, characterize, and generalize sepsis endotypes in diverse populations to facilitate new and better prognostic and therapeutic solutions, especially in low- and middle-income countries that carry the greatest burden for sepsis[15]. Toward this goal, we analyze host gene expression in a large prospective multi-site international sepsis cohort in West Africa, Southeast Asia, and the United States as part of the Austere environments Consortium for Enhanced Sepsis Outcomes (ACESO)[10,11]. We employ soft-clustering decomposition that allows for the visualization and identification of both discrete and overlapping clusters within high-dimensional gene expression data, thereby accommodating the syndromic nature of sepsis[16]. As a result, we can report the identification of molecular endotypes from the combined global cohort that predict sepsis mortality. Interpretation of the new gene expression features described here from understudied populations suggests that immune dysregulation is a key component of the biology linked to sepsis heterogeneity and outcomes.

## Methods
### Study sites and subjects
Blair P.W. et al.[17] describe a detailed description of the entire study cohort. Five hundred and six patients across three sites had specimens available for this study. Study protocols were approved by the Naval Medical Research Command (NMRC) Institutional Review Board (IRB) (Cambodia sepsis study # NMRC.2013.0019; Ghana sepsis study # NMRC.2016.0004-GHA; Duke sepsis study (Duke # PRO00054849) in compliance with all applicable Federal regulations governing the protection of human subjects as well as host country IRBs. The study protocol in Cambodia was approved by the Cambodian National Ethics Committee for Health Research (NECHR). The protocol in Ghana was approved by the Committee on Human Research, Publication, and Ethics (CHRPE) at Kwame Nkrumah University of Science & Technology. All procedures were in accordance with the ethical standards of the Helsinki Declaration of the World Medical Association. All patients, or their legally authorized representatives, provided written informed consent.

### Demographic comparisons
To compare gender, size, age, and mortality in the study, we considered all the patients with available 28-day mortality information. For continuous variables (age) were compared using Welch's two-sample t-test, while discrete variables (mortality) used the Wilcoxon rank sum test[18]. To show the distribution of individual data, medians, and interquartile ranges, the age was plotted and colored in relation to gender, mortality, and site using functionalities of ggplot[19], and ggridges[20] packages in R statistical environment[21].

### RNA-seq library preparation from patient peripheral blood
Specific details for the preparation of RNA for sequencing are described in Rozo et al.[22]. Briefly, the peripheral blood RNA was collected in PAXgene RNA tubes (PreAnalytiX), total RNA was purified using PAXgene Blood miRNA KIT (Qiagen) and depleted of human rRNA and globin using Globin-Zero Gold rRNA Removal Kit (Illumina). The vendor (Azenta) prepared the paired-end RNA sequencing according to standard procedures and sequenced generating ~ 50 million, 150-bp long paired-end reads per sample.

### Genome alignment and sequencing data pre-processing
Sequencing data were aligned to the human genome (GRCh38). Briefly, all samples underwent quality control using *FastQC*[23]. Passing samples were aligned to the genome using *Hisat2*[24], and transcripts were assembled using *Stringtie*[25]. Low-expression features (counts per million <10), sex-linked features (located on chromosomes X, and Y), and features not mapping to known genes were removed to decrease noise and avoid gender bias in feature selection and modeling. To normalize the data between different study sites, the raw read counts were normalized using Median Ratio Normalization (MoR) method[26]. Normalized data were then transformed using Variance Stabilizing Transformation (VST). The processing yielded 3061 genes for 506 patients from across the three study sites.

### Dimensionality reduction
To aid in data interpretation we employed multiple dimensionality reduction methods including principal component analysis (PCA)[21], uniform manifold approximation, and projection (UMAP)[27]. For PCA and UMAP, data were filtered to contain the 494 subjects with 28-day mortality information and analyzed in an R statistical environment[21]. Topological data analysis (TDA) was used to group participants with comparable gene expression profiles in an unbiased manner[28,29] and identify trends and endotypes within the sepsis cohort. TDA was performed using the AyasdiAI machine intelligence platform (Symphony AyasdiAI, Palo Alto, CA, USA) on the normalized RNA sequencing data, employing a correlation metric combined with two proprietary *neighborhood* lenses. Groups of patients with similar gene expression profiles were defined within the TDA structure based on node density and connectivity. By its very essence, TDA captures the continuous nature of data[30], and it therefore allows for a degree of overlap between adjacent phenotypes (i.e., a patient can be a member of more than one group at the same time). In the TDA network presented here, resolution and gain settings were selected that limited the degree of overlap between groups (less than 10% of participants were allowed to be shared), while at the same time maintaining the integrity of the TDA structure. PCA and UMAP analysis was plotted using *ggplot2*[19] package in R[21]. All figures were finalized using Adobe Illustrator.

### Differential gene analysis within and *between* TDA groups
The normalized expression table of 3061 genes for 505 patients was filtered to only contain 494 patients with known mortality outcomes. In case of *between* TDA groups comparison, patients that could be attributed to multiple TDA groups were removed leaving only patients exclusive to individual TDA groups. *Within* TDA groups, patients were compared by 28-day mortality, contrasting those who died by 28-day and those who lived to day 28. The significance of the mean difference was evaluated using a Welch two-sample t-test and the final p.value was adjusted for multiple comparisons using Benjamini & Hochberg[31] model in R[21]. The complete results of both analyses can be found in the corresponding tables in Supplementary Data 2. Data were sorted by p.adjusted or p-value along with $\log_2$ fold change (L2FC) to highlight the most significantly changed genes. Terms for all comparisons were combined and respective L2FC values were plotted as a heatmap using the *ggplot2*[19] package in R[21]. To cluster genes, the distance matrix was calculated using *manhattan* method[32], row order was determined by unsupervised hierarchical clustering using *mcquitty* method[33] and the dendrogram was plotted using the *ggdendro* package[34] in R. Resulting figures were finalized using Adobe Illustrator.

### Gene set enrichment analyses within and *between* TDA groups
Gene set enrichment analysis was performed using the results of differential gene expression analyses *within* and *between* TDA groups. For each contrast, the log2 fold changes (L2FC) were calculated using group means and ranked in decreasing order. For the 'between' TDA comparison, only genes with significant p.adjust value ($\leq = 0.05$) were considered for the analysis. All genes were considered for the *within* TDA comparison due to the low count of statistically significant values. For gene set enrichment analysis of the entire cohort comparing 28-day mortality, we used genes with significant

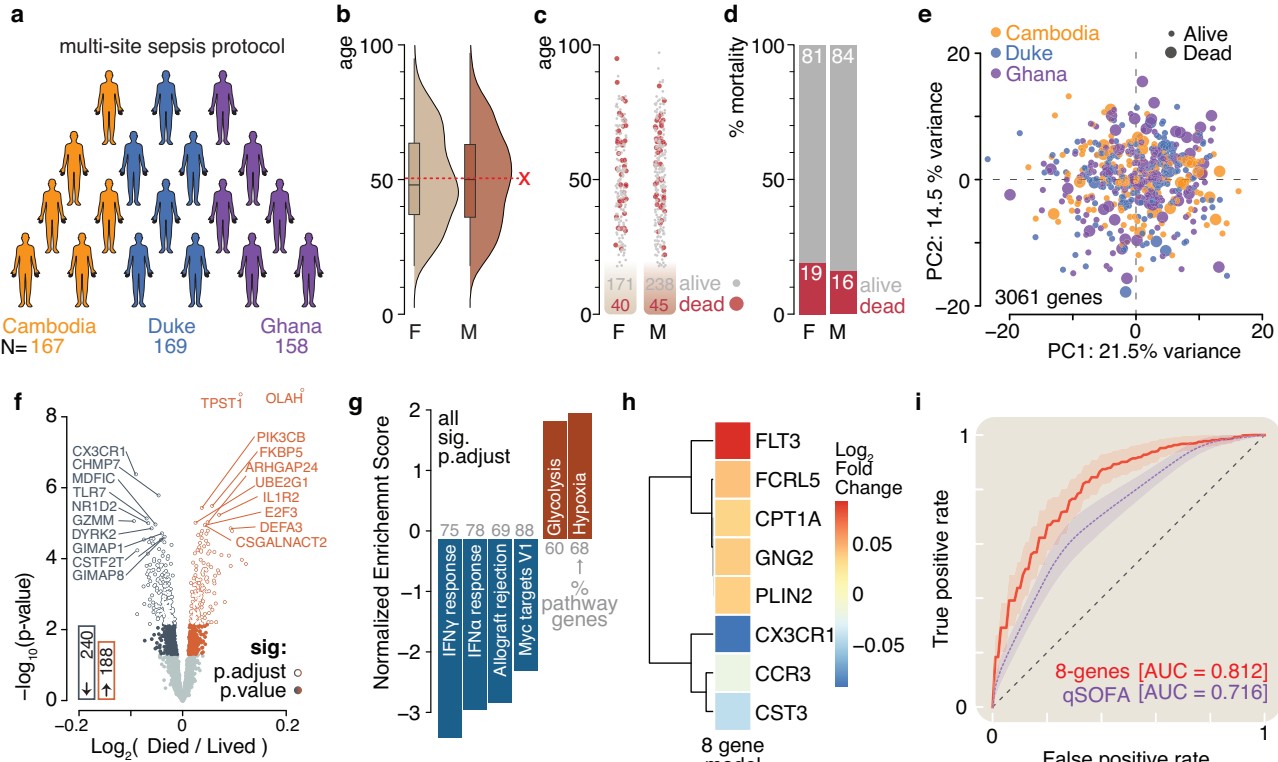

**Fig. 1 | Experimental design, demographics, and initial prognostic modeling.** **a** Data for this study was generated from subjects enrolled in three different sites located in Cambodia (orange), Duke (blue) (USA), and Ghana (purple). **b** The mean and median age were nearly identical (~50 years of age) for each gender (F: female – light tan; M: male – light brown). The median and quartiles are indicated by boxplot and the mean as red dotted lines. **c, d** The 28-day mortality was also very similarly distributed between genders and averaged a ~ 17% death ratio for the entire study cohort. Red spheres and bar fill indicate patients that died. **e** Principal component analysis of the 3061 selected genes did not reveal any site-specific or mortality-specific patient clustering. Point color indicates site, and point size indicates mortality. **f** Volcano plot of differential gene expression analysis comparing subjects that died by day 28 to those that survived. Labels show for top ten most significantly

changed genes (p.adjust ≤ 0.05). The bar insert shows a total number of significantly changed genes (p.adjust ≤ 0.05, blue – decrease, red – increase). **g** Results of gene set enrichment analysis using Molecular Signature Database Hallmark gene data sets show significantly different pathways (p.adjust ≤ 0.05). Numbers in gray show the percentage of pathway coverage. **h** The 28-day mortality-based $\log_2$ fold changes of eight genes were used for prognostic mortality modeling. **i** Receiver operating characteristic curve showing performance of eight-gene model (AUC = 0.812 ± 0.070) in predicting 28-day mortality relative to the predictive power of quick sepsis-related organ failure assessment (qSOFA, AUC = 0.716 ± 0.075) score. The lightly shaded regions surrounding the curves correspond to 0.5 standard deviations. All modeled curves were generated with repeated stratified k-fold cross-validation described in the methods.

p.value (≤=0.05). The selected genes were compared against the Molecular Signature Database gene set representing Hallmark pathways accessed using the *msigdbr*[35] package and analyzed using the *clusterProfiler*[36] package. For the full cohort, we set *pvalueCutoff* parameter to 0.05 to detect significant pathways only, whereas for other analyses we set this parameter to 1 and marked the statistically significantly enriched pathways in the figures with boxes for a clearer comparison. All the gene set enrichment results are shown in Supplementary Data 2. Final data was plotted as stacked barplot or heatmap using *ggplot2*[19], rows organized by unsupervised clustering using the *manhattan* distance matrix and the *ward.d* clustering method[37], and a dendrogram plotted using *ggdendro* package[34] in R. Resulting figures were finalized using Adobe Illustrator.

### Performance modeling
All modeling as well as plotting of ROC curves were performed in Python (v3.9.12). The *LogisticRegression*, *RepeatedStratifiedKFold*, and *cross_val_score* functions from the *sklearn (v1.1.2)*[38] python package were used to perform modeling. The logistic regression was run with a *max_iter* setting set to 1000. The stratified k-folds were run with 10 repeats and 10 splits. The cross-validation score used *roc_auc* as a scoring metric. The mean and standard deviation of the resulting AUROCs were calculated using *mean* and *std* functions from the *sklearn*[38] python package. ROC curves were plotted using *RocCurveDisplay* and *pyplot* functions from *sklearn* and

*matplotlib (v3.5.1)*[39] python packages respectively. Resulting figures were finalized using Adobe Illustrator.

### Comparison of the clinical parameters
All clinical comparative analysis was conducted in the R statistical environment[21]. The data were filtered to include 480 patients used in the TDA analysis. The available clinical features were filtered to those collected at timepoint 0 (T00h) and those for which measurements were collected single time and were unlikely to change throughout the study (i.e., medical history). The 28-day mortality time point was also included, due to its study relevance. The features were further filtered to those with data available for at least two patients in each TDA group being compared. To compare groups the patients were combined into a single group if needed (i.e., t1 + t2 + t3 = t123). The variance was evaluated using *aov*, and TDA group comparisons were evaluated using *t.test* built-in R functions respectively[21]. The results of the analyses are provided in Supplementary Data 1.

### Statistics and reproducibility
All comparisons were evaluated using standard statistical methods built-in into the R statistical computing environment. The different TDA comparison patient groups were clearly defined and are described in Supplementary Fig. 2d. All statistical significance tests were subject to multiple testing corrections and the adjusted p-value (p.adjusted) was reported first,

**Fig. 2 | Topological Data Analysis of the combined global sepsis cohort.** The normalized expression of the top 1000 genes from 506 subjects was used to perform Topological Data Analysis (TDA). This decomposition method groups patients based on their gene expression profiles and produces a relationship network consisting of nodes (clusters of patients with similar gene expression) and edges (one or more patients are shared between two adjacent nodes). **a** The sepsis TDA network was divided into four groups along a left-to-right axis, which corresponded to differences in 28-day (d28) mortality. The TDA network was colored by 28-day mortality on a continuous scale ranging from green (0% of patients in the node died) to red (40% or more of patients in the node died). **b** Patients were stratified into three high-mortality groups (t1, t2, t3) and one low-mortality group (t4-5) (yellow - highlights TDA subgroup; red – indicates mortality).

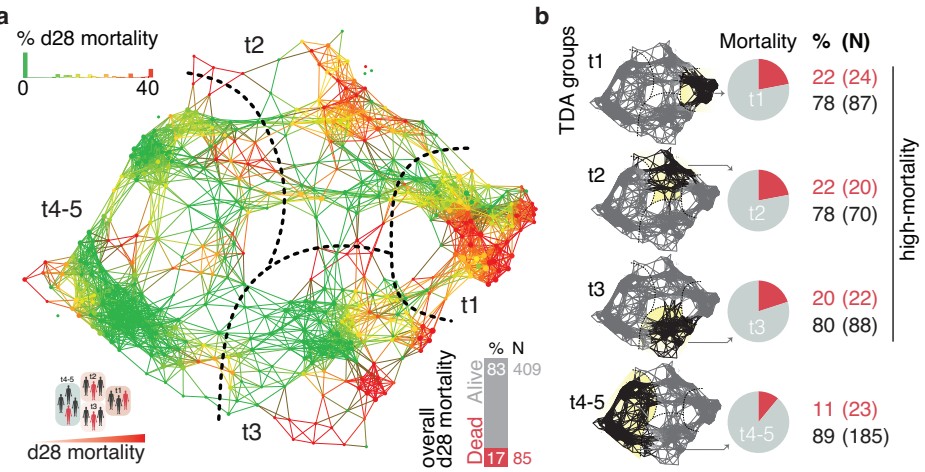

**Fig. 3 | TDA group-specific genes have increased performance versus a clinical tool.** The TDA group-specific feature selection identified a total of thirteen genes for stratified 28-day mortality prognostic. **a** Heatmap of hierarchically clustered genes in 28-day (d28) mortality comparison across the entire study cohort showing TDA group membership and direction of expression change. **b** TDA overlay showing expression of two of the biomarker genes across the entire study cohort. TDA groups are indicated with dashed lines and labels. **c** Receiver operating characteristic curves showing the performance of prognostic classifiers including the eight-gene model (red) based on the entire cohort versus quick sepsis-related organ failure assessment (qSOFA) score (purple), and TDA-stratified models (blue). Lightly shaded areas surrounding the curves correspond to 0.5 standard deviations. All modeled curves were generated with repeated stratified k-fold cross-validation described in the methods.

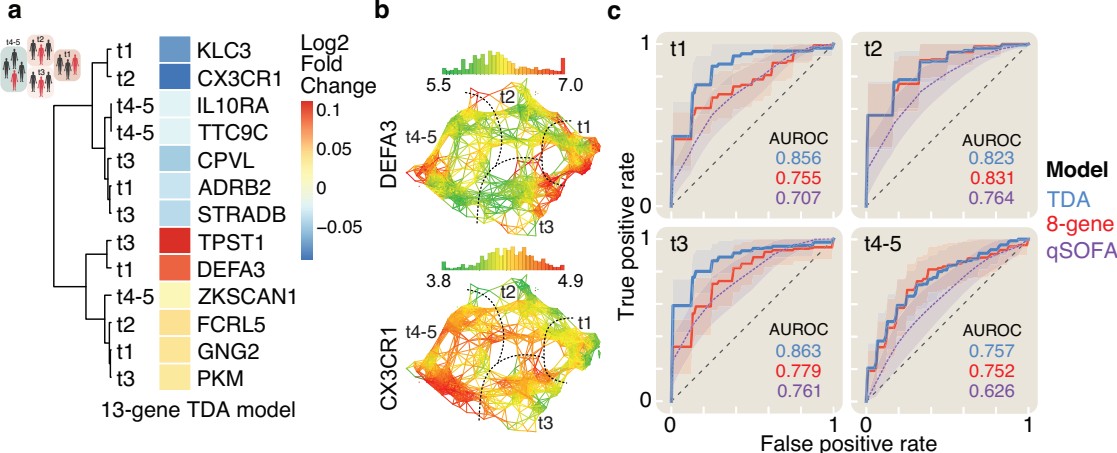

followed by the p-value second in case corrected values were insignificant in order to reflect trends. This study was based on the results and materials collected in previously completed human clinical trials that did not involve sample replicate collection.

## Reporting summary

Further information on research design is available in the Nature Portfolio Reporting Summary linked to this article.

## Results

### Gene expression predicts mortality in global sepsis cohorts

Total RNA from peripheral blood collected at enrollment (July 2016 – October 2017) was sequenced from 494 subjects in observational studies of sepsis in Ghana, Cambodia, and Durham, North Carolina (Fig. 1a, Supplementary Fig. 1a)[17,22,40]. Male and female subjects were similar in age and 28-day mortality in the combined cohort (Fig. 1b, Supplementary Fig. 1b). The overall 28-day mortality, defined as the percentage of patients that died within 28 days of enrollment into the study, was 17%, with individual cohorts having 28-day mortalities of 8% (Duke), 13% (Cambodia), and 32% (Ghana), respectively (Figs. 1c and 1d, Supplementary Fig. 1c, and 1d). Inspection of the data following preprocessing revealed no site bias (Fig. 1e,

Supplementary Fig. 1e, 1f, 1g, 1h). Differentially expressed genes between 28-day survivors and non-survivors include transcripts previously linked to sepsis outcomes such as *IL1R2, OLAH*, and *CX3CR1* (Fig. 1f and Supplementary Fig. 1i and 1j)[6,41]. Gene set enrichment analysis[42,43] was used to identify biological pathways enriched in non-survivors. Hypoxia was the top enriched term while interferon response was significantly reduced (Fig. 1g). Feature selection using the Minimum Redundancy Maximum Relevance[44,45] algorithm was performed using 391 differentially expressed genes to develop a prognostic classifier for 28-day mortality (Supplementary Data 1). The use of Minimum Redundancy Maximum Relevance resulted in a list of the top 50 ranked genes in decreasing relation to the risk of sepsis-related mortality (Supplementary Data 1). Average performance versus the number of features used in logistic regression was determined and resulted in eight transcripts (Fig. 1h). These top eight were used as inputs to logistic regression. Performance with qSOFA scores that combine mental status, respiratory rate, and blood pressure was also evaluated in these same subjects[1]. The eight-transcript model had improved performance (AUROC = .81) versus the model using qSOFA alone (AUROC = 0.72) in the 441 subjects where matched data were available (Fig. 1i). These results show that a molecular prognostic for sepsis mortality can be derived from diverse global cohorts.

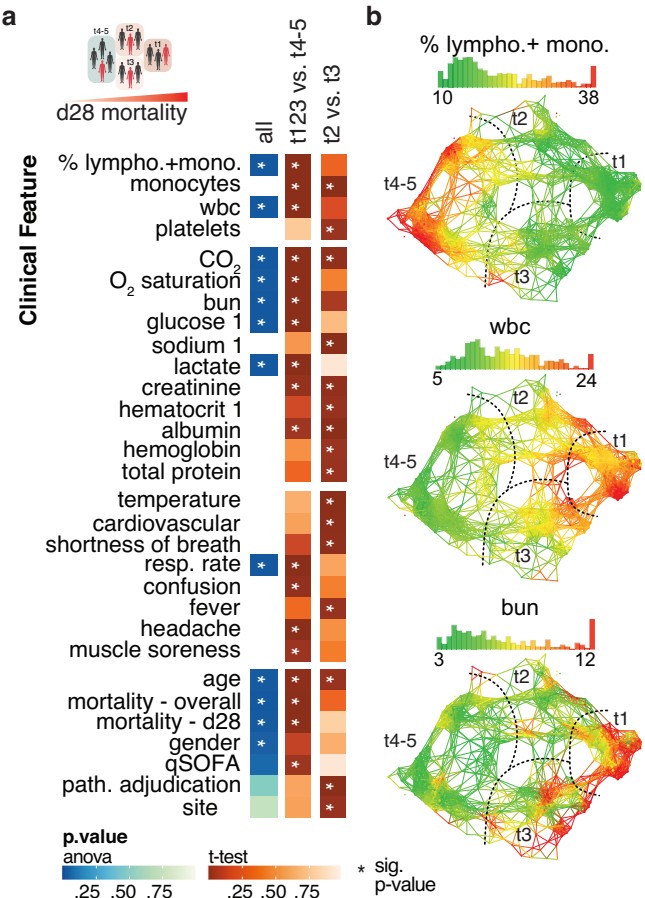

**Fig. 4 | Clinical features distinguish stratified TDA groups. a** Diverse clinical features were evaluated across the TDA network from the combined global cohort plotted as a heatmap. The analysis of variance and statistical group comparisons identify important clinical variables across the TDA network (blue) and between different TDA groups (orange-red) respectively. Only selected clinical parameters recorded for more than >60% of subjects are shown. The white star depicts the significant p-value (≤0.05). **b** TDA overlays show the distribution of candidate clinical parameters across the cohort and TDA groups in accordance with statistical results. Red and green indicate the maximum and minimum values respectively, while yellow represents the middle of the gradient.

## Data dimension reduction identifies patient groups that improve prognostic performance

Genome-wide expression profiling has been used successfully to identify septic shock subclasses that describe biological heterogeneity[5,7]. We hypothesized that heterogeneity within, and across global cohorts could compromise the performance of the molecular sepsis prognostic. To identify relevant groups for improved prognostic models, we used topological data analysis (TDA) to cluster patients in an unsupervised, data-driven manner[16] (Fig. 2a). In TDA, a 2-dimensional topological network is created based on similarities between data points, as well as the overall distribution of the data in n-dimensional space (Supplementary Fig. 2a). This provides an intuitive means of stratifying subjects with similar gene expression profiles into groups where relative position within the network can reveal shared biology or endotypes. Nodes in the TDA network correspond to groups of patients that are clustered together based on similarities in their gene expression, with the node size dependent on the number of patients in each node. Since TDA captures the continuous nature of data, it allows for a degree of overlap between adjacent nodes, meaning that patients can be included in more than one node at a time. This is represented by the edges in the network, which indicate that one or more patients are shared between two nodes[46] (Supplementary Fig. 2a).

TDA analysis of the gene expression data resulted in the identification of 5 patient groups along a major left-to-right axis (Fig. 2b & Supplementary Fig. 2b). Further analysis showed that this axis corresponded to differences in 28-day mortality of the patients: TDA groups t1, t2, and t3 all exhibited elevated rates of 28-day mortality (22%, 22%, and 20% respectively) versus the *low mortality* groups t4 and t5 (10% and 11%, respectively) (Fig. 2b and Supplementary Fig. 2b). Feature selection was repeated within the TDA groups to ask if 28-day mortality prognostic performance could be improved using a stratified approach. Groups t4 and t5 contained too few non-survivors individually for this analysis, but shared a subset of subjects and were therefore combined into one low mortality group t4-5 (28-day mortality=11%) (Supplementary Fig. 2b and c). A total of 13 features across these four final groups were identified by step-up performance analysis of the previously identified 50 Minimum Redundancy Maximum Relevance transcripts (Fig. 3a and b, Supplementary Fig. 3, Supplementary Data 1). Models consisting of features for specific TDA groups were fitted and evaluated using only the subjects belonging to the respective TDA group. Performance increased in each of the 3 high-mortality groups (t1,2,3), and was decreased in the low-mortality t4-5 group versus what was achieved for the combined global cohort (Figs. 1i and 3c). All these group-specific features and models proved to be more effective than qSOFA (Fig. 3c).

## Clinical and laboratory features of high-mortality patient groups describe multi-organ dysfunction

We have previously shown multiple clinical and physiological measures that are abnormal in this combined global cohort including elevated white blood cells, blood urea nitrogen, and lactate[17]. To define the clinical basis of the TDA-stratified patient groups we asked if physiologic and laboratory features varied across the entire network and within the two major axes of the network. Chemistry and hematology features including lactate, blood urea nitrogen, lymphocytes, and white blood cells varied significantly across the network (Fig. 4a and Supplementary Data 2). We looked more specifically at the left-right axis described by the low (t4-5) and high-mortality (t1,2,3) groups for significant differences. We noted a dramatic contrast in lymphocyte-monocyte percentage (up in low mortality) versus white blood cells (up in high mortality) (Fig. 4a and b and Supplementary Data 2). Lactate, blood urea nitrogen, and qSOFA are also significantly elevated in the high mortality groups while oxygen saturation is decreased (Fig. 4a, b, Supplementary Fig. 4, Supplementary Data 2). Additional symptoms including confusion in the combined high-mortality groups indicate multi-organ dysfunction in these subjects. Inspection of the vertical axis defined by groups t2 and t3 revealed a site-specific bias including pathogen etiologies and hemoglobin\hematocrit (Fig. 4a and Supplementary Data 2). This is consistent with our previous findings that show adjudicated bacterial, viral, fungal, and parasitic infections were significantly different across the Ghana, Cambodia, and U.S. populations[11].

## Molecular definition of sepsis endotypes

Gene set enrichment analysis using molecular features was used to explore the TDA-stratified groups. We first compared gene set enrichment analysis results across the major left-right mortality axis (Fig. 5a, b, and Supplementary Data 2). Type I and II interferon responses and allograft rejection were elevated in the low mortality t4-5 group, as were pathways linked to cell growth and proliferation including MYC and E2F targets. Key interferon-stimulated genes including *IFI27* and *ISG15* were significantly elevated in this group (Supplementary Fig. 5). These signatures are consistent with an active adaptive immune response in the low-mortality patient group t4-5. In contrast, the high-mortality groups were enriched for terms including hypoxia and coagulation which are characteristic of severe sepsis with group t1 being the most elevated. The individual gene-level inspection identified genes such as *CYP1B1*, *S100A12*, and *IL1R2* that were most elevated in this high-mortality group t1 (Supplementary Fig. 5 and Supplementary Data 2). Notably, these genes characterize a CD14[+] immunosuppressive monocytic myeloid-derived suppressor cell population MS1 recently described in bacterial sepsis and COVID-19 as negatively associated with survival[47].

**Fig. 5 | Gene-set enrichment comparison of TDA groups and 28-day mortality. a** Heatmap of normalized enrichment values (NES) for the Molecular Signature Database Hallmark pathways comparing TDA groups to each other. The pathways were grouped by unsupervised hierarchical clustering. **b** Stacked barplot of NES values for the Molecular Signature Database Hallmark pathways comparing patients by 28-day (d28) mortality within TDA groups. Black boxes with star in Fig. 5a and b indicate statistically significant enrichment in gene set enrichment analysis (p.adjust ≤ 0.05). Supplementary Fig. 2d describes the number of patients used in each comparison (dead/alive and pair-wise columns). To perform gene set enrichment analysis against the Molecular Signature Database Hallmark gene data set, for Fig. 5a we used genes with significant $\log_2$ fold change values (p.adjust ≤ 0.05), and for Fig. 5b all genes. The TDA groups were colored and shaded as follows: t4-5 – blue; t3 – yellow; t2 - orange; t1 – red.

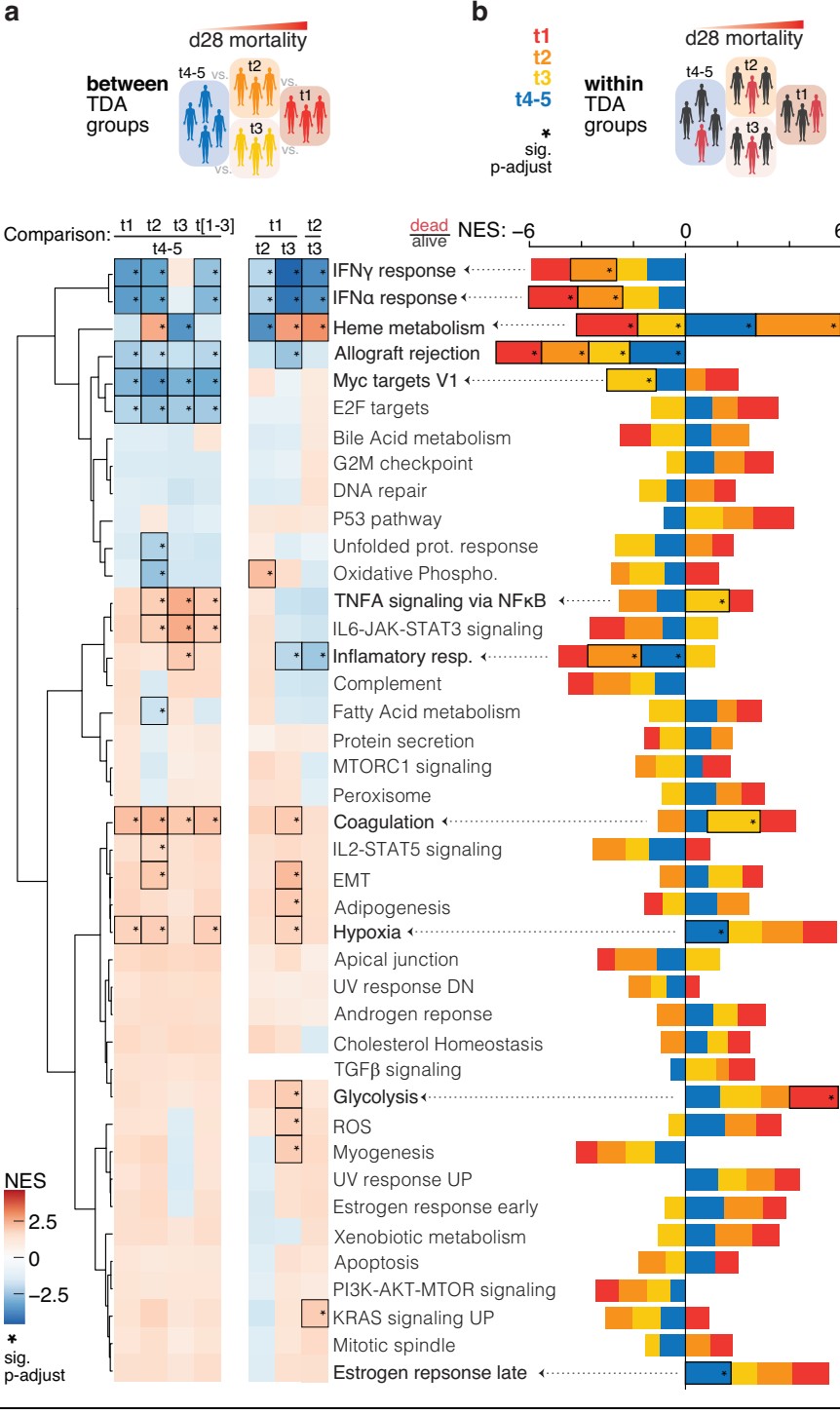

Patient group t1 showed the least interferon response signatures by gene set enrichment analysis versus all other groups. Taken together these data are consistent with an immunosuppressed phenotype in group t1. Pathways linked to inflammation including TNFα and IL6-JAK-STAT3 signaling were also elevated in the high mortality groups compared to the low mortality group. Comparison across the three high mortality groups revealed that group t3 is most enriched for acute inflammatory signatures. We assessed differences between groups t2 and t3 and noted a significant difference in heme metabolism consistent with the clinical hemoglobin \hematocrit laboratory results (Fig. 4a and Supplementary Data 2). We also asked if we could identify pathways linked to mortality within the potential endotypes. A subset of signatures including interferon response and allograft rejection describing the adaptive immune response was positively

linked to survival in all groups (Fig. 5b). However, we did note pathways with TDA group-specific differences by outcome including the inflammatory response term that was significantly enriched in survivors in the immunocompetent low-mortality group t4-5.

## Discussion

TDA decomposition of host gene expression from our combined global sepsis cohort suggests at least 4 endotypes (Fig. 6). The low mortality t4-5 group with less severe clinical correlates is enriched for immunocompetent molecular and hematological signatures most notably for the adaptive immune response. This is in direct contrast to high mortality group t1 which has reduced lymphocytes and interferon gene expression signatures and has molecular markers consistent with immunosuppressive cells.

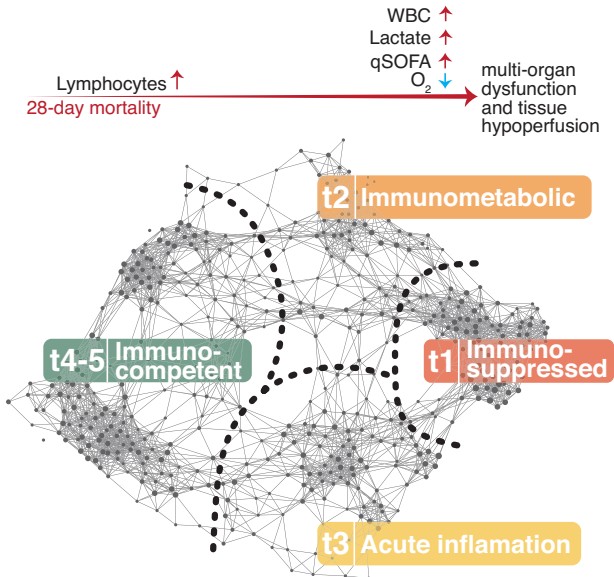

**Fig. 6 | Sepsis endotype model.** TDA network related to outcomes including 28-day mortality and clinical and molecular features observed in different stratified patient groups. Red horizontal arrow indicates 28-day mortality across the TDA network. Small vertical arrows indicate direction of change of specific analyte. Red color indicates increase, while blue indicates decrease.

High-mortality group t3 shares key clinical features of severe illness with group t1, but gene set enrichment analysis shows that there is a robust innate and adaptive immune response in this group. Group t2 is notable for heme metabolism signatures and reduced mitochondrial gene expression linked to oxidative phosphorylation. Overall, results from our combined global cohort are similar to recent reports of biological endotypes using unsupervised analysis of clinical and multi-omics data from sepsis studies. Sweeney and colleagues identified three stable patient groups in a combined analysis of bacterial sepsis cohorts designated Inflammopathic, Adaptive, and Coagulopathic[8]. Another study found three stable sepsis subclasses coined immune-innate, immune-coagulant, and immune-adaptive[9]. Finally, Scicluna and colleagues identified four endotypes in European cohorts designated Molecular Diagnosis and Risk Stratification of Sepsis (Mars) 1-4[6]. The Mars1 group was characterized by reduced expression of T-cell and adaptive immune genes versus a Mars3 group with robust innate immune activation. Consistent across all these studies is the association of reduced mortality and less severe clinical correlates with endotypes most strongly defined by signatures of a functional adaptive immune response similar to our *immunocompetent* group t4-5. In contrast, subclasses in these studies that have reduced adaptive immune signatures, coagulopathies, elevated mortality, and poor clinical scores align with our *immunosuppressed* group t1. The t3 group described here, with innate pathway signatures linked to *acute inflammation* and high mortality shares these features with the *inflammopathic* group reported by Sweeney and colleagues[8]. Finally, group t2 characterized by heme biosynthesis is consistent with the described *immunometabolic* endotypes and free heme has been shown to contribute to sepsis pathogenesis[6,48].

This study reinforces the generality of common sepsis endotypes, introduces to the best of our knowledge new cohorts of relatively understudied populations from West Africa and Southeast Asia, and identifies potential new therapeutic approaches to improving sepsis outcomes. The endotype-specific gene expression features that we identify are similarly regulated in stratified patient groups in multiple sepsis studies published over the past decade (Supplementary Fig. 6)[7,49–51]. Notably, there is great interest in using endotype and biomarker-guided strategies for precision medicine interventions that incorporate immunotherapeutics in subjects predicted to benefit[52]. Lymphocyte growth factors including IL7 have shown promise in promoting the innate immune response in humans[53].

Accordingly, endotypes have been proposed to contribute to inconclusive results from immunomodulator trials in sepsis subjects[10]. A direct link between outcomes and endotypes has been shown following the analysis of data from patients treated with hydrocortisone[9,10]. Corticosteroid treatment was shown to have differential effects on survival in subjects based on immunophenotypic endotypes. Interestingly, our data offers some preliminary support to this finding when we consider gene expression signatures in survivors and non-survivors within TDA groups. We note that terms including TNFα and IL6 signaling, and inflammatory response are increased in survivors in the *immunocompetent* group t4-5 but decreased in survivors in the *acute-inflammation* group t3 (Fig. 5b). Future studies will validate our endotypes and prognostic models in additional international sites across the globe[40].

Conducting complex observational studies in low- and middle-income countries presents unique challenges but is necessary to create a more complete definition of sepsis endotypes and can only be done through a global study using unified methods. Pathogen identification is especially limiting because the infecting pathogen is often not identified in the developing world. In the United States, common causative bacterial pathogens include *E. coli*, *Staphylococcus aureus*, and *Streptococcus pneumoniae*[54]. In Africa, common sepsis etiologies include *Salmonella spp.*, malaria, and *Mycobacterium tuberculosis*, particularly in persons living with HIV infection[55]. We have previously shown that sepsis screening tools that are widely used during clinical care had a sub-optimal performance for risk stratification in diverse international cohorts[17]. Pathogen diversity and the syndromic nature of sepsis might contribute to this result and encourage future studies to reveal endotypes linked to pathogens endemic to low- and middle-income countries or parasitic co-infections.

## Data availability

Raw sequence data and processed subject gene level data used in this study have been deidentified and deposited in dbGaP (Accession: phs003661.v1) under restricted access in compliance with study informed consent and National Institutes of Health Human Subjects Protection guidelines. Data can be obtained following local IRB approval and with a letter of collaboration with the primary study investigator(s). Source data underlying the data analysis used to generate the figures in this study can be found in Supplementary Data 1 and 2.

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

## Acknowledgements

We thank the Naval Medical Research Command Genomics group at the Biological Defense Research Directorate as well as Carlo Colantuoni and Derren Barken for their support. This work was accomplished through the efforts of many in the Austere environments Consortium for Enhanced

Sepsis Outcomes. We also thank the clinical research teams at Komfo Anokye Teaching Hospital in Ghana, Duke University Hospital in North Carolina, and Takeo Provincial Hospital in Cambodia. This work was supported by the Defense Health Agency through the Joint West Africa Research Group, Combating Antibiotic Resistance Bacteria (FY1819 0130.1832), and Defense Threat Reduction Agency (JSTO-CBA) to Naval Medical Research Command (NMRC) (HDTRA1516108), Defense Health Bureau of Medicine & Surgery to NMRC for Naval Medical Logistics Command Cooperative Agreement (N626451920001). K.L.S. is an employee of the US government and A.G.L. is a military service member. This work was prepared as a part of official duties. Title 17 U.S.C. 105 provides that Copyright protection under this title is not available for any work of the United States Government. Title 17 U.S.C. 101 defines a U.S. Government work as a work prepared by a military service member or employee of the U.S. Government as part of a person's official duties. The views expressed reflect the results of research conducted by the authors and do not necessarily reflect the official policy or position of the Henry M. Jackson Foundation for the Advancement of Military Medicine, Inc., Department of the Navy, Department of Defense, nor the United States Government.

## Author contributions

J.G.C., D.V.C., K.L.S., C.W.W., and E.L.T. conceived the study. Gene expression analysis and bioinformatics was performed by P.G., J.B., D.A.S. G.B.F and J.G.C. Feature selection and prognostic modeling was performed by G.B.F. and E.C. Data interpretation was performed by J.G.C., D.V.C., P.G., J.B., D.A.S., E.D., I.B., and S.K. Clinical interpretation was contributed by P.W.B., C.O., A.O.O, C.G.B., G.O., T.V., and A.G.L. The manuscript was written by J.G.C., P.G., J.B., G.B.F. and D.A.S. with input from all authors.

## Competing interests

The authors declare the following competing interests: D.V.C., J.B., J.G.C., and D.A.S. are listed as inventors on a U.S. Provisional Application No. 63/578,492, entitled: "Prediction of Mortality of Patients Diagnosed with Sepsis," applied for by the Henry M. Jackson Foundation for the Advancement of Military Medicine, Inc., on the identification of molecular endotypes that predict sepsis mortality. E.L.T is an employee and has equity in Danaher. E.L.T and C.W.W have received consulting fees from Biomeme. All other authors have no competing interests to declare.
