## [Peer Review File · Communications Medicine]

Reviewers' comments:

Reviewer #1 (Remarks to the Author):

The purpose of this study was to develop a method based upon transcriptomic changes in blood to be able to stratify patients with sepsis into groups that will determine their risk of poor outcomes. In addition, the study team hoped to discover molecular mechanisms that were responsible for favorable vs poor outcomes in sepsis. The study has a number of strengths including the large number of patients, the state-of-the-art analytical tools that were employed for analysis, and the diverse patient population. Furthermore, the manuscript is well-written and the findings are very interesting and of potential significance for future therapeutic development. There are a few points that should be addressed however.

Specific comments

1. The concept of the analysis is fairly straight forward but I don't understand the TDA plots themselves. I think that each vertex is a particular patient, and that the axes are supposed to represent reductions of many components into two dimensions to facilitate viewing. But if each vertex is an individual patient why are they connected as though it's depicting a network? Shouldn't each vertex be instead a single data point with each of them plotted, as they would, based on the two dimensions? The authors should explain the methodology in a more clear cut fashion.
2. The TDA plots indicate that mortality increases to the right, as though it is the ordinate of the axis, this is confusing as mortality is a binary event. Additionally, there is already an indicator of mortality in the coloring, why does one plot require 2 indicators of mortality.
3. For figure 2B, the left part of the panel is confusing and unnecessary. The coloring uses the same colors that indicate mortality in the same part of the figure, additionally the groups are defined as a part of the rest of the figure.

Reviewer #2 (Remarks to the Author):

The investigators have contributed a important dataset of significant sample size for the field, with their cohort strengthened by accrual at multiple sites. They identify "molecular endotypes" in whole blood RNA-seq from sepsis patients (N~500). The subcohorts differ greatly in mortality rates (8% to 32%). The authors generate an 8 transcript model to predict mortality (AUROC = 0.8). Next, subgroups with higher (3 subgroups) or lower mortality (2 combined into 1 subgroup) were identified, with some site-specific differences. The molecular signatures of these groups were associated with several gene expression patterns (e.g., high mortality-hypoxia/coag; IFN/allograft rejection GSEA-low mortality).

Validation: The validity of the conclusions are strengthened by the fact it reflects similar conceptual findings as similar studies (e.g., MARS, SRS). However, a major limitation of the work is the lack of a validation cohort for their findings, some of which focus on the "predictive power" of their 8-/13-gene sets. While reproducing their analysis in another cohort (e.g., public blood transcriptomic dataset) is likely outside of the scope of this work, some sort of analysis or commentary that more rigorously compares/contrasts their "immunocompetent" and "immunodysregulated" subcohorts or the activity of their 8 or 13-gene sets in other cohorts may be helpful.

Neutrophils: Neutrophils may contribute significantly to the whole blood transcriptomic dataset. While the authors may have tested the % and abs neutrophil association to the “endotypes” (e.g., when analyzed data for Figure 4), it would be helpful if they described the results.

Hallmark: I understand that the project was not designed as a mechanistic study. However, for Figure 5 (GSEA) it may be helpful to show in the supplement what specific genes were driving the enrichment of key gene sets (e.g., perhaps some of the ones highlighted in text or the figure).

Time of sample collection: Blair et al. 2023 detail characteristics of the cohort. Blair et al. 2023 endorse that blood samples were obtained at time of enrollment. However, the time from hospitalization to enrollment/sample collection are not clear and are critically important for a transcriptomic study. Also did the time post-hospitalization to sample collection vary by site?

--Edy Kim MD PhD
Brigham and Women's Hospital / Harvard Medical School

22 AUG 2023

It is a pleasure to submit our revised manuscript COMMSMED-23-0375-A titled “*Sepsis Endotypes Identified by Host Gene Expression Across Global Cohorts.*” Sepsis is recognized as a health priority by the World Health Organization, however, globally inclusive studies that consider low, and middle-income countries with the greatest sepsis burden are rare. Our study fills this gap and contributes a more complete definition of sepsis endotypes to drive the development of targeted therapies. Please find below our point-by-point responses to reviewer comments. We appreciate the constructive suggestions by both reviewers.

Reviewers' comments:

Reviewer #1 (Remarks to the Author):

The purpose of this study was to develop a method based upon transcriptomic changes in blood to be able to stratify patients with sepsis into groups that will determine their risk of poor outcomes. In addition, the study team hoped to discover molecular mechanisms that were responsible for favorable vs poor outcomes in sepsis. The study has a number of strengths including the large number of patients, the state-of-the-art analytical tools that were employed for analysis, and the diverse patient population. Furthermore, the manuscript is well-written and the findings are very interesting and of potential significance for future therapeutic development. There are a few points that should be addressed however.

We appreciate the reviewer’s thoughtful summary of our work and study strengths including diverse populations. The highest sepsis burdens have been identified in sub-Saharan Africa and southeast Asia (Rudd 2020). We anticipate that the inclusion of understudied populations in our study will fill a much-needed research gap and will be of great interest to the field. As pointed out by the reviewer we have employed “state-of-the-art” analytical approaches with the specific goal of catalyzing new therapeutic approaches that consider sepsis heterogeneity.

Specific comments

1. The concept of the analysis is fairly straight forward but I don’t understand the TDA plots themselves. I think that each vertex is a particular patient, and that the axes are supposed to represent reductions of many components into two dimensions to facilitate viewing. But if each vertex is an individual patient why are they connected as though it’s depicting a network? Shouldn’t each vertex be instead a single data point with each of them plotted, as they would, based on the two dimensions? The authors should explain the methodology in a more clear cut fashion.

We thank the reviewer for the suggestion to clarify the TDA plot methodology and interpretation. Each “vertex” that results from the Topological Data Analysis is actually a group of participants that share common gene expression profiles and not a single participant. These are formally called “nodes” and appear as circles in the plot. The size of the “node” reflects the number of participants that have this particular gene expression profile, with larger nodes comprising more participants. Direct connections between two nodes are termed “edges,” which indicates that a subset of subjects is shared between two nodes. Although each individual node could be considered a gene expression “endotype” in its own right, we follow a more customary approach in which clusters of highly interconnected nodes within the

network are considered to represent an endotype. Hence, our TDA patient groups t1 to t5 comprise multiple adjacent nodes with shared edges. We have added language to the manuscript text and Figure 2 legend to make this clearer. In addition, a graphic in supplemental Figure S2A has been updated to highlight a few important features of the TDA plots.

2. The TDA plots indicate that mortality increases to the right, as though it is the ordinate of the axis, this is confusing as mortality is a binary event. Additionally, there is already an indicator of mortality in the coloring, why does one plot require 2 indicators of mortality.

As indicated above we recognize the value of explaining the TDA in a more clear-cut fashion and have added additional text to the results section, figure legends, as well as updated Figure 2 and Supplemental Figure 2 to achieve this.

With respect to mortality within TDA network: each of the nodes (vertex\circles) in the TDA plot represents multiple subjects. Mortality is calculated as a percentage of dying subjects within a particular node. The mortality bar plot in Figure 2A (top left) reflects the coloring of nodes based on this mortality calculation. The mortality depiction here is not a binary but a node frequency or fraction. The specifics of how TDA nodes and relationships are made are now better explained in supplemental Figure S2A.

As the reviewer noted, the percent mortality between four TDA groups indeed increases from left to right or along the ordinate axis. We repositioned the TDA group cartoon next to the gradient-filled triangle to make this relationship clearer in Figure 2A (bottom left).

Finally, we added a definition of “28d mortality” to the start of the Results section.

3. For figure 2B, the left part of the panel is confusing and unnecessary. The coloring uses the same colors that indicate mortality in the same part of the figure, additionally the groups are defined as a part of the rest of the figure.

We thank you reviewer for the comment. We would like to retain this panel as it gives an overview of TDA group locations within the network and helps establish the left-to-right symmetry inherent/apparent in this TDA depiction. To address the reviewer’s comment, and make this point clearer, we changed panels to grayscale to avoid reference to mortality, highlighted individual TDA groups with yellow, and linked them to Figure 2B (with small arrows) depicting the TDA group's specific mortality.

Reviewer #2 (Remarks to the Author):

The investigators have contributed a important dataset of significant sample size for the field, with their cohort strengthened by accrual at multiple sites. They identify “molecular endotypes” in whole blood RNA-seq from sepsis patients (N~500). The subcohorts differ greatly in mortality rates (8% to 32%). The authors generate an 8 transcript model to predict mortality (AUROC = 0.8). Next, subgroups with higher (3 subgroups) or lower mortality (2 combined into 1 subgroup) were identified, with some site-specific

differences. The molecular signatures of these groups were associated with several gene expression patterns (e.g., high mortality–hypoxia/coag; IFN/allograft rejection GSEA–low mortality).

We are excited that the reviewer recognizes the value of a large molecular dataset generated from a global sepsis cohort. We previously reported challenges with commonly used clinical screening tools when applied to diverse populations (Blair 2023). Beyond our work reported here, we anticipate that our data will promote the generation and validation of robust endotypes that can be generalized to any population.

Validation: The validity of the conclusions are strengthened by the fact it reflects similar conceptual findings as similar studies (e.g., MARS, SRS). However, a major limitation of the work is the lack of a validation cohort for their findings, some of which focus on the “predictive power” of their 8-/13-gene sets. While reproducing their analysis in another cohort (e.g., public blood transcriptomic dataset) is likely outside of the scope of this work, some sort of analysis or commentary that more rigorously compares/contrasts their “immunocompetent” and “immunodysregulated” subcohorts or the activity of their 8 or 13-gene sets in other cohorts may be helpful.

We acknowledge that the validation cohort is important to further support our findings and thank the reviewer for recognizing that reproducing all our analysis in such a cohort would be beyond the scope of this work. Nevertheless, to address reviewer comments, fill the validation gap, and further support our 8- and 13-gene sets we analyzed their activity in new and older landmark studies published in the sepsis field. We extracted expression information for genes from our 8- and 13-gene sets, then performed differential gene comparison in line with the findings of the published work, and visually highlighted the agreement with our study findings. The results (Supplementary Figure S6.) show that the gene expression behavior of our sets of genes is highly similar (nearly identical) and reproducible in diverse sepsis cohort data sets and data types (microarray, expression array, RNA-Sequencing, and single-cell sequencing). Furthermore, similar to our study, the analyzed cohorts describe either cell types, endotypes, or sepsis patient groups that can be classified as immune-competent (lower mortality or lesser symptoms) and immune-dysregulated (higher mortality or symptom severity). The fact that our 8- and 13-gene sets behave in the same fashion across the sepsis cohort data published over the past decade supports our findings and is in line with our endotype descriptions. We hope that the reviewer will find this analysis sufficient as validation of our findings.

Neutrophils: Neutrophils may contribute significantly to the whole blood transcriptomic dataset. While the authors may have tested the % and abs neutrophil association to the “endotypes” (e.g., when analyzed data for Figure 4), it would be helpful if they described the results.

In line with the reviewer’s comment, the added value of sequencing nucleic acid from whole peripheral blood is that all blood cells will be analyzed. Indeed, neutrophils differ across the network (ANOVA significant), and are enriched in high mortality groups 1/2/3 vs 4-5 (t-test <.05), however, since the number of neutrophil hematology measures is below the threshold we set of 60% of all subjects included in our RNA sequencing dataset, we did not include it in the figure. All these data are now provided in Supplementary Data 2 table along with the number of subjects with given measurements.

Hallmark: I understand that the project was not designed as a mechanistic study. However, for Figure 5 (GSEA) it may be helpful to show in the supplement what specific genes were driving the enrichment of key gene sets (e.g., perhaps some of the ones highlighted in text or the figure).

We would like to bring the reviewers' attention to Supplementary Data Table 2, which includes all the data relevant to the GSEA analysis. Specifically included is a list of enriched genes (sometimes referred to as leading edge genes) that are responsible for the enrichment of each gene set described in Figure 5. We apologize for not including this information in our earlier submission.

Time of sample collection: Blair et al. 2023 detail characteristics of the cohort. Blair et al. 2023 endorse that blood samples were obtained at time of enrollment. However, the time from hospitalization to enrollment/sample collection are not clear and are critically important for a transcriptomic study. Also did the time post-hospitalization to sample collection vary by site?

We agree with the reviewer that time-after-onset/hospitalization is an important factor(s) in mechanistic studies, and should be appropriately constrained in order to advance our understanding of sepsis biology and disease pathways. However, as with demographics, medical history, and pathogen type, duration is one of those factors that commonly contributes to sepsis heterogeneity. Our aim for this study was to describe gene expression endotypes that would be routinely encountered in the clinical setting, irrespective of location, infection type, symptom duration, and so on. As such, we recruited a wide variety of patients with the objective of describing commonalities (in gene expression) across a very heterogeneous population. The practical advantage of this is that our findings, including the prognostic biomarker panels, are generalizable to "sepsis" as a whole. As pointed out in the paper, many of the genes identified by us were found to be relevant in prior studies of sepsis, by other groups, and in other (often less heterogeneous) study populations. This lends strength to our findings and their applicability for patient stratification and prognostication. However, further refinements to clinical protocols and enrollment criteria are undoubtedly warranted, and our observations will form the basis for future mechanistic studies, where time-after-onset/hospitalization will be accounted for in study design.

--Edy Kim MD PhD
Brigham and Women's Hospital / Harvard Medical School

REVIEWERS' COMMENTS:

Reviewer #1 (Remarks to the Author):

The authors have done a nice job of responding to my concerns and the manuscript is much improved.

Very nice addition to the field.

Reviewer #2 (Remarks to the Author):

I appreciate the effort that the authors put into this revision, both to clarify figures and the eFigure 6 with comparison to other prominent sepsis datasets. As in my initial review, I appreciate the authors' contribution of an important clin dataset and analysis.

Major comment:

I may have missed an eMethods file, but the authors should write supplemental methods describing the generation of Suppl Figure 6.

Edy Kim

Brigham & Women's / Harvard Med Sch

We thank the reviewers for their review of our resubmitted manuscript titled, “Sepsis Endotypes Identified by Host Gene Expression Across Global Cohorts.” Please find below a point-by-point response to the provided comments.

REVIEWERS' COMMENTS:

Reviewer #1 (Remarks to the Author):

The authors have done a nice job of responding to my concerns and the manuscript is much improved.

Very nice addition to the field.

We appreciate the comments provided by the reviewer throughout this peer review process. They have strengthened the manuscript, and we are excited to share our results with the field.

Reviewer #2 (Remarks to the Author):

I appreciate the effort that the authors put into this revision, both to clarify figures and the eFigure 6 with comparison to other prominent sepsis datasets. As in my initial review, I appreciate the authors' contribution of an important clin dataset and analysis.

We thank the reviewer for their critical review and insightful comments in the initial review. These suggestions have greatly improved our manuscript.

Major comment:

I may have missed an eMethods file, but the authors should write supplemental methods describing the generation of Suppl Figure 6.

We have now provided the supplemental methods as requested.

Edy Kim
Brigham & Women's / Harvard Med Sch